# High-Sensitivity Sensing in All-Dielectric Metasurface Driven by Quasi-Bound States in the Continuum

**DOI:** 10.3390/nano13030505

**Published:** 2023-01-27

**Authors:** Zhao Jing, Wang Jiaxian, Gao Lizhen, Qiu Weibin

**Affiliations:** 1Computer Science and Information Engineering School, Xiamen Institute of Technology, Xiamen 361021, China; 2College of Information Science and Engineering, Huaqiao University, Xiamen 361021, China

**Keywords:** all-dielectric metasurface, Fano resonance, torodial dipole, bound state in the continuum, refractive index sensing

## Abstract

Quasi-bound states in the continuum (quasi-BIC) in all-dielectric metasurfaces provide a crucial platform for sensing due to its ability to enhance strong matter interactions between light-waves and analytes. In this study, a novel high-sensitivity all-dielectric sensor composed of a periodic array of silicon (Si) plates with square nanoholes in the continuous near-infrared band is theoretically proposed. By adjusting the position of the square nanohole, the symmetry-protected BIC and Friedrich–Wintgen BIC (FW–BIC) can be excited. The torodial dipole (TD) and electric quadruple (EQ) are demonstrated to play a dominating role in the resonant modes by near-field analysis and multipole decomposition. The results show that the sensitivity, the Q-factor, and the corresponding figure of merit (FOM) can simultaneously reach 399 nm/RIU (RIU is refractive index unit), 4959, and 1281, respectively. Compared with other complex nanostructures, the proposed metasurface is more feasible and practical, which may open up an avenue for the development of ultrasensitive sensors.

## 1. Introduction

Metasurface is an artificially arranged array of periodic subwavelength structures with unusual electromagnetic characteristics, such as negative refractive index [1], near-zero refractive index [2], and negative magnetic permeability [3]. With the development of micro-nano processing technology, metasurfaces are widely used in metalens [4], light absorbers [5], biochemical sensors [6], and optical modulators [7]. Compared with three-dimensional (3-D) structures, photonic crystals, or metal resonators, dielectric supersurfaces with high Q-factor have advantages in terms of flexibility, size, and nonlinear optical compatibility [8,9,10]. The all-dielectric metasurface supports Mie resonance and can provide high Q-factor and FOM (Figure of Merit), and is more compatible with complementary metal oxide semiconductor production processes [10,11,12]. Therefore, all-dielectric metasurface-based sensors exhibit stronger light-harvesting capabilities and smaller volumes, enabling label-free, on-chip integration, and ultrasensitive sensing, which become current research hotspots. In 2020, Jeeyoon et al. presented a metasurface structure composed of an all-dielectric hollow cuboid array, which can be used for refractive index sensing with a maximum sensitivity of 161 nm/RIU and a FOM of 78 [13]. In 2021, Zhang et al. suggested a “lucky junction”-shaped all-dielectric nanostructure that is insensitive to polarization and incident angle, with a sensitivity of 986 nm/RIU and a FOM of 32.7 [14]. In 2022, Song et al. proposed a refractive index sensor composed of two asymmetric rectangular hollow silicon cylinders, with a maximum sensitivity of 160 nm/RIU and a FOM of 575 [15].

The idea of bound states in the continuum (BIC) has been included into various metasurface designs in order to achieve higher Q resonances [16,17,18,19,20,21]. BIC designates a state that still preserves locality in the continuous domain, which can be explained by destructive interference. The coupling of the BIC resonant mode to all radiated waves vanishes when the system parameters are continuously changed, leading to an infinite lifetime, which denotes an infinitely high Q value, making it impossible to directly observe the BIC resonance from the scattering spectrum, also known as the dark mode [22,23]. There are two main methods for implementing BICs in practical design [17]. One approach is to construct a symmetry-protected BIC, which prevents the leaking of the bound state by embedding a bound state with a specific symmetry inside a continuum with a different symmetry. The second technique is to produce an “accidental” BIC by modifying the target system’s parameters in order to cancel the continuous output wave [24,25]. Breaking the symmetry to transform the BIC into a quasi-BIC, which has a shape resembling that of Fano, can be used in practice to generate the leaking resonance [26,27]. Studies have demonstrated that the metal and dielectric symmetry-protected BIC metasurfaces can support quasi-BIC resonance by breaking the symmetry or oblique incident [22,26,28,29].

There are numerous different designs available at the moment to achieve high-Q quasi-BIC mode. For instance, two nanoblocks of different thickness [30], length [31], or width [32], and different arrangements of polymer nanoblocks [33,34], typically achieve resonance by the near-field interaction between the inter-cell gaps, and the presence of internal gaps makes them more sensitive to manufacturing errors. A single asymmetric unit has been the subject of some investigations in order to create Fano resonance. This single structure [35,36] has a bigger process error, since the coupling between the cells is not taken into account. Therefore, researchers are interested in exciting Fano resonance in a single resonator configuration. Recently, there has been an urgent need to create on-chip integrated sensing and detection chips for applications such as medical testing, environmental monitoring, food safety, and others that call for portability and speedy on-site inspection [37]. The interaction between light and matter can be significantly improved by the nano-optical resonance structure, which also offers multi-dimensional light field management capabilities in the space domain and frequency domain. It may further realize the fully chip-integrated optical sensing and detecting functions and be readily integrated with numerous functionalities [10]. The silicon material device technology is established, affordable, and compatible with the CMOS process [38,39]. Thus, researchers’ interest has been drawn to silicon-based high refractive index sensor chips.

In this study, we propose an all-dielectric metasurface refractive index sensor composed of asymmetric square nanohole arrays that exhibit TD and EQ resonance modes in the near-infrared region. The symmetry-protected BIC and FW–BIC, which correspond to infinite Q-factor, can be excited by adjusting the placement of the nanohole. Furthermore, the distribution of surface current and field further supports the excitation of BIC-driven TD and EQ resonance. At the same time, the structural parameters were optimized to obtain a more sensitive sensor, and the spectral response of the refractive index of the medium was calculated. The maximum sensitivity obtained was 399 nm/RIU, the quality factor was 4959, and the FOM was 1281. The metasurface sensor device designed in this paper provides a theoretical reference for the realization of ultrasensitive high-refractive-index micro-nano sensor design.

## 2. Structural Design

The biosensing platform based on the all-dielectric square nanoholes metasurface is shown in Figure 1. The metasurface structure is formed by a periodic array of silicon (Si) plates with a thickness of 170 nm and nanoholes located on silicon dioxide (SiO_2_) substrate. Figure 1a shows the schematic diagram of the sensing device of the metasurface. Figure 1b,c represent the unit and top view of the metasurface structure, respectively. The period of the structure is *P_x_* = 1100 nm and *P_y_* = 990 nm, the side length of the Si plate is *W*_1_ = 400 nm and *L* = 800 nm, and the side length of the square nanohole is *W*_2_ = 170 nm, respectively. The center distance between the nanohole and the Si plate along the y-axis is represented by Δ. When Δ is equal to 0, the metasurface structure is completely symmetric, and when Δ is not equal to 0, the in-plane symmetry of the structure is broken. Here we only consider the movement of the nanohole along the negative y axis, and Δ is defined as the asymmetric parameter. We numerically simulated the spectral response of the metasurface using the finite element method, where Floquet periodic boundary conditions are applied in the x, y directions, the incident field is an x-polarized plane wave propagating along the negative z axis, and the refractive index of Si is 3.5, and the refractive index of the SiO_2_ substrate is set to 1.45.

## 3. Results and Discussion

### 3.1. Resonance Performance Analysis

The transmission spectrum for various symmetry parameters Δ is shown in Figure 2a. For mode 1, the resonance peak vanishes at Δ = 0 nm, indicating there is no energy leakage from the bound state to the free space continuum state, and the corresponding radiation Q-factor tends to infinity. However, at Δ ≠ 0 nm, the resonance peak is still visible. A sharp asymmetric linear shape, the Fano resonance, is shown in Figure 2b at Δ = 10 nm. We can observe from Figure 2a, as Δ increases, that the resonance peak of mode 1 becomes wider. This proves that the BIC resonance mode of the excited device is a symmetry-protected BIC mode by breaking the symmetry of the structure. When Δ ≠ 0 nm, the in-plane symmetry of the structural unit is perturbed, and the BIC mode transforms into a quasi-BIC mode that can radiate and have a high Q-factor due to the establishment of a radiative channel between the nonradiative bound state and the free-space continuum state, resulting in more incident light being radiated into the free space. When the translation distance is smaller, the radiation channel is narrower, the radiation loss energy is smaller, and the Q-factor is larger. For mode 2, when Δ = 0 nm, there is resonance peak. Figure 2c depicts the Fano resonance, the sharp asymmetric linear shape, at Δ = 10 nm. The resonance peak of mode 2 gets increasingly narrower as Δ increases. The resonance peak disappears at Δ = 48 nm, corresponding to an accidental BIC that can be explained by destructive interference between resonance modes.

High-Q metasurfaces need more sophisticated manufacturing techniques because their near-field coupling is so susceptible to manufacturing flaws. It is typically more difficult to construct metasurface devices, making it challenging to produce ultra-high-Q metasurface resonators in experiments. The wider the line width of the quasi-BIC mode, the smaller the Q factor.

We can fit the transmission spectra of modes 1 and 2 with the typical Fano formula [40,41]
(1)$$T=T_{0}+A_{0}\frac{{\left[q+2(\omega -\omega_{0})/\Gamma \right]}^{2}}{1+{\left(2(\omega -\omega_{0})/\Gamma \right)}^{2}}$$
where *ω_0_* represents the resonance peak frequency, Г represents the resonance attenuation, which is proportional to the resonant peak line width, *T*_0_ is background scattering parameter, that is, the electromagnetic wave without any interaction, corresponds to the continuous state, *A*_0_ is the coupling coefficient for the continuous state and the discrete state, *q* is the Breit–Wigner–Fano parameter, which determines the asymmetry of the resonance spectrum.

For asymmetric linear Fano resonances in transmission spectra, we usually determine the radiative *Q*-factor by fitting the spectra to Equation (1). The *Q*-factor is calculated using the following formula: $Q=\omega_{0}/\Gamma$. Figure 2b,c show the results of the transmission spectrum curve fitting of the mode 1 and mode 2 at Δ = 10 nm. The resonance wavelength of mode 1 is 1420 nm, and the corresponding *Q*-factor is 2450, and the resonance wavelength of mode 2 is 1433, and the corresponding *Q*-factor is 2488.

We also investigated the connection between the asymmetry parameter and Q-factor of mode 2, as shown in Figure 3. The continuous state interacts with the bound state and leaks when the structure’s symmetry is disrupted, producing ultra-high Q Fano resonance [26]. Figure 3 shows that Q-factor and the asymmetry parameter meet the quadratic reciprocal relationship Q∝Δ^−2^, and it can be demonstrated that this mode is a symmetry-protected BIC mode [22,26].

Furthermore, to visualize the electromagnetic excitation modes corresponding to the lowest values of transmittance, we present a schematic diagram of the near-field distribution of the cell structure at the resonant wavelength position, as shown in Figure 4. The field patterns are normalized to the incident electromagnetic field, and the maximum electric field enhancement and magnetic field distribution can be up to 140- and 350-fold, respectively. There are similar field distributions at the resonance wavelength of 1420 nm and 1433 nm. In Figure 4a, the vortex arrows inside the unit structure represent displacement current loops with opposing rotational directions in xoy plane, which excites opposite MDs on the z-axis, and in Figure 4b, the vortex arrows within the structure represent the production of magnetic loops in the yoz plane. Combining the incident light with the x polarization, the circular magnetic ring on the yoz plane excites a strong magnetic TD along the x-axis. At the same time, a pair of reverse rotating currents will also excite the magnetic quadrupole, which has a strong contribution to the resonance response of the metasurface. The displacement current distribution in the structural gap in the xoy plane is an electric quadrupole, as shown in Figure 4a. However, we can only observe the dominant electromagnetic contribution near each resonance in the near-field pattern.

In order to study the characteristics of Fano resonance, we use the multipole decomposition method in the Cartesian coordinate system to calculate the contribution of each multipole to the resonance response. Here, we only consider electric dipoles P, magnetic dipoles M, and toroidal dipoles T. Electrons, electric quadrupoles $Q^{\left(e\right)}$, and magnetic quadrupoles $Q^{\left(m\right)}$ can be ignored due to the small contribution of higher-order multipoles. The multipole moment is defined as [42,43,44]
(2)$$P=\frac{1}{i\omega }\int jd^{3}r,$$
(3)$$M=\frac{1}{2c}\int \left[r\times j\right]d^{3}r,$$
(4)$$T=\frac{1}{10c}\int \left[\left(r\cdot j\right)r-2r^{2}j\right])d^{3}r,$$
(5)$$Q_{\alpha ,\beta }^{\left(e\right)}=\frac{1}{2i\omega }\int \left[\left(r_{\alpha }j_{\beta }+r_{\beta }j_{\alpha }\right)-\frac{2}{3}\delta_{\alpha ,\beta }\left(r\cdot j\right)\right])d^{3}r,$$
(6)$$Q_{\alpha ,\beta }^{\left(m\right)}=\frac{1}{3c}\int [{\left(r\times j\right)}_{\alpha }r_{\beta }+{\left(r\times j\right)}_{\beta }r_{\alpha }]d^{3}r,$$
where ***j*** is the current density vector, ***r*** is the position vector, *c* is the speed of light, *ω* is the angular frequency, *α*, *β* = *x*, *y*, *z*. The scattered power for multipole momentum can be calculated by
(7)$$I_{P}=\frac{2\omega^{4}}{3c^{3}}{\left|P\right|}^{2},$$
(8)$$I_{M}=\frac{2\omega^{4}}{3c^{3}}{\left|M\right|}^{2},$$
(9)$$I_{T}=\frac{2\omega^{6}}{3c^{5}}{\left|T\right|}^{2},$$
(10)$$I_{Q^{\left(e\right)}}=\frac{\omega^{6}}{5c^{5}}\sum {\left|Q_{\alpha \beta }^{\left(e\right)}\right|}^{2},$$
(11)$$I_{Q^{\left(m\right)}}=\frac{\omega^{6}}{40c^{5}}\sum {\left|Q_{\alpha \beta }^{\left(m\right)}\right|}^{2},$$

The multipole decomposition calculation results in logarithmic coordinates are shown in Figure 5. We can observe that the maximum far-field radiation is mainly contributed by the electric quadrupole and the toroidal dipole at the resonance wavelength of 1420 nm. At the resonance wavelength of 1433 nm, the electric quadrupole dominates the resonance, and the toroidal dipole contributes secondarily. Additionally, the resonance response of the metasurface is significantly influenced by the magnetic quadrupole. The result of the multipole decomposition is consistent with the near-field distribution in the Figure 4.

### 3.2. Influence of Structural Parameters on Transmission Performance

The calculated transmittance spectra are shown in Figure 6 when Δ changes from 0 nm to 48 nm. As marked by the circles in Figure 6, the radiation Q-factor of mode 2 tends to infinity at Δ = 0 nm, which proves the existence of symmetry-protected BICs. Continuing increasing Δ, the in-plane symmetry of the unit cell is perturbed, allowing the BIC modes to transform to quasi-BIC modes that are radiable and have high Q value and mode 1 appear and become wider. For mode 1, the parameter-tuned BIC is realized by adjusting Δ until the Friedrich–Wintgen condition is met.

We analyzed the effect of structural parameter changes on the transmission spectrum when Δ = 10 nm, as shown in Figure 7. It is clear that the resonance wavelength is more sensitive to the height and side length of the Si plate and the square nanohole than it is to the period. With the period *P* increasing, there is a little redshift in the resonance transmission spectrum of the metasurface. As the Si plate’s height *H* and side lengths *W*_1_ and *L* grow, the resonance becomes noticeably red-shifted. However, the resonance exhibits a distinct blue shift as the square nanohole’s side length rises. It can be explained by the effective refractive index of the metasurface. As the side length of the nanohole increases, the effective refractive index of the metasurface declines, and the interaction between light and matter in the resonant cavity gradually reduces, resulting in a blue shift of the resonance position. As the period, side length of the Si plate increase, the surface area of the metasurface, resulting in an increase in the effective refractive index, and the interaction between light and matter in the resonant cavity gradually increases, resulting in a red shift of the resonance position [45].

The resonant line width and *Q*-factor, however, are little affected by an increase in period H and side length *W*_1_. Additionally, with an increase in side length *L* from 780 nm to 820 nm, the *Q*-factor of mode 1 goes from 814 to 14,447 whereas the *Q*-factor of mode 2 goes from 2268 to 4026. As side length *W_2_* increases from 150 nm to 190 nm, the *Q*-factor of mode 1 first decreases from 10,347 to 2420, then increases to 5733, whereas the *Q*-factor of mode 2 increases from 708 to 12,539. While the *Q*-factor of mode 2 reduces from 7918 to 1552 with an increase in height H from 150 nm to 190, the *Q*-factor of mode 1 increases from 1239 to 13,091.

### 3.3. Analysis of Refractive Index Sensing Characteristics

The higher the sensitivity of the sensor, the easier it is to detect small changes in the dielectric properties around the sensor. The sensitivity can be expressed as [46] S=Δλ/Δn, where Δλ is the variation of the resonance wavelength, Δn is the variation of the refractive index caused by the change of the surrounding environment, and the unit is RIU (Refractive Index Unit). The entire performance of the sensor can be described by its FOM, and a higher FOM value denotes a better performance of the sensor. The FOM can be represented as [47] FOM=S/FHWM, where the FWHM is the width at half-peak of the Fano resonance line. A high Q-factor (i.e., narrow line width) indicates that the sensor has strong light-field confinement capabilities, whereas a high sensitivity sensor indicates that more light field leaks from the device into the space. As a result, sensitivity and Q factor have a mutually exclusive relationship that influences the FOM.

Figure 8a,b illustrate the impact of external media with various refractive indices n on the transmission spectrum. The liquid to be tested with n from 1.31 to 1.35 fills the gap and around the structure, and the thickness of the liquid is 300 nm, and the asymmetric parameter Δ is 30 nm. The increasing of refractive index has little effect on the Q-factor and the peak value of the resonance peaks, and the resonance position exhibits a noticeable redshift. The corresponding sensitivities of each resonance are 282 nm/RIU and 399 nm/RIU, respectively. The related Q factors are 1688 and 4959, while the corresponding FOMs are 308 and 1281, respectively. The field distribution is primarily responsible for the variation in sensing performance between the two resonance modes. Reducing structural asymmetry can further boost the device’s sensitivity and Q factor, which will further enhance FOM. Compared to metal or hybrid metal-dielectric sensors, the FOM are higher. There are certain advantages in sensing performance when compared to some previously proposed architectures [13,14,41]. As shown in Table 1, after comparing our results with previous works, it can be seen that the proposed structure has better sensing performance and simpler structure, providing a valuable reference for future sensor applications.

## 4. Conclusions

In this paper, an all-dielectric metasurface refractive index sensing platform based on TD and EQ resonance driven by quasi-BIC is proposed and demonstrated theoretically. The resonance in the transmission spectrum evolves from symmetry-protected BIC into quasi-BIC and from quasi-BIC into FW–BIC by translating the nanohole. The physical basis of TD and EQ resonance can be shown by the near-field distribution and multipole decomposition. By analyzing the effect of metasurface on the sensing performance, it was discovered that the maximum sensitivity of the sensor can reach 399 nm/RIU, the quality factor is 4959, and the FOM is 1281. Higher Q values can be obtained with symmetry-protected BIC, and FOM can be further enhanced by lowering the asymmetry parameter. Additionally, by adjusting various structural characteristics, the Fano resonance can be modified to be more advantageous for sensing applications. The proposed metasurface structure is less complicated and more useful than existing complex nanostructures, offering a theoretical benchmark for biochemical sensing platforms.

## Figures and Tables

**Figure 1 nanomaterials-13-00505-f001:**
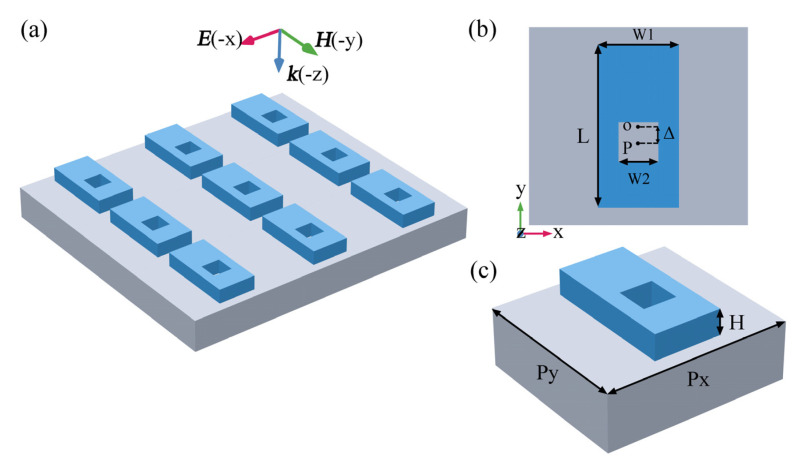
Schematic diagram of the structure of an all−dielectric metasurface. (**a**) Schematic diagram of the overall outline of the asymmetric nanohole metasurface; (**b**) cell structure diagram of the metasurface; (**c**) top view of the cell structure.

**Figure 2 nanomaterials-13-00505-f002:**
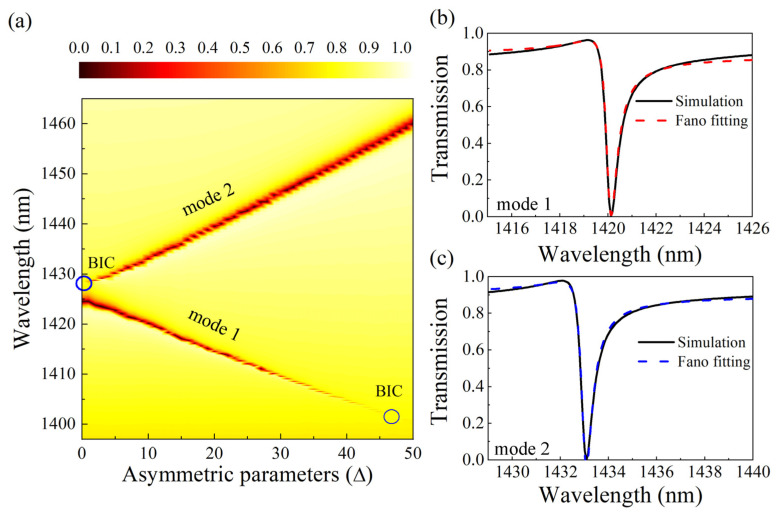
Metasurface transmission spectrum analysis. (**a**) Transmission spectra of metasurfaces under different asymmetric parameters; (**b**,**c**) Fano fitting of transmission spectra of mode 1 and mode 2, respectively, at Δ = 10 nm; the solid line represents the simulation result, and the dashed line represents the fitting result of the Fano formula.

**Figure 3 nanomaterials-13-00505-f003:**
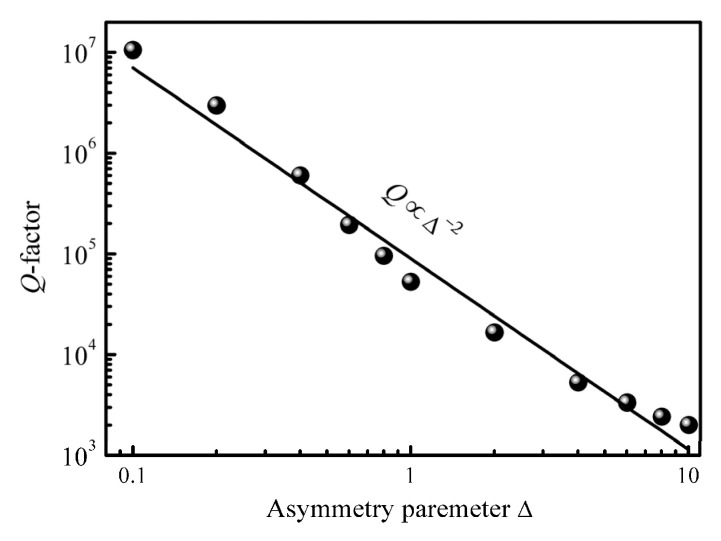
Plot of Q-factor of mode 2 as a function of the asymmetric parameter Δ (log-log scale), where the black points are obtained by the simulation method and the black line is fitted to demonstrate the inverse quadratic dependence of Δ.

**Figure 4 nanomaterials-13-00505-f004:**
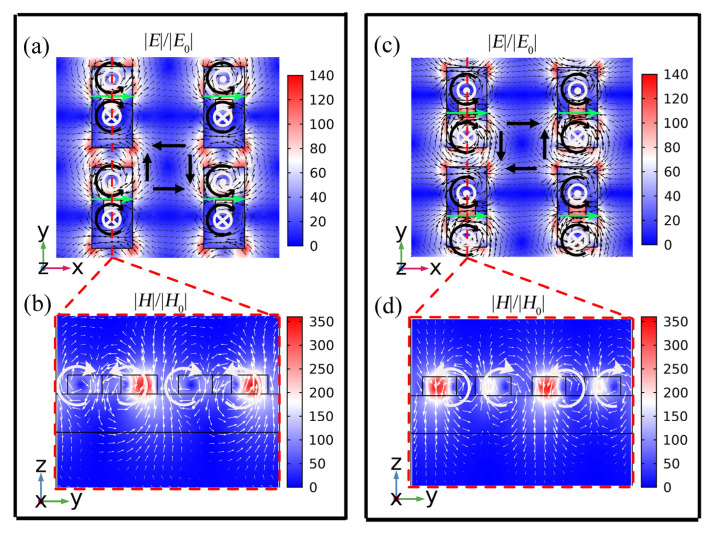
Field distribution at resonant wavelength at Δ = 10 nm (**a**,**c**) the normalized electric field distribution of the resonant wavelength of mode 1 and mode 2, respectively, in the xoy plane; (**b**,**d**) the normalized magnetic field distribution of the resonant wavelength of mode 1 and mode 2, respectively, in the yoz plane. The black arrows represent the displacement current direction, and the white arrows represent the magnetic field direction, and the green arrows represent the toroidal dipoles direction.

**Figure 5 nanomaterials-13-00505-f005:**
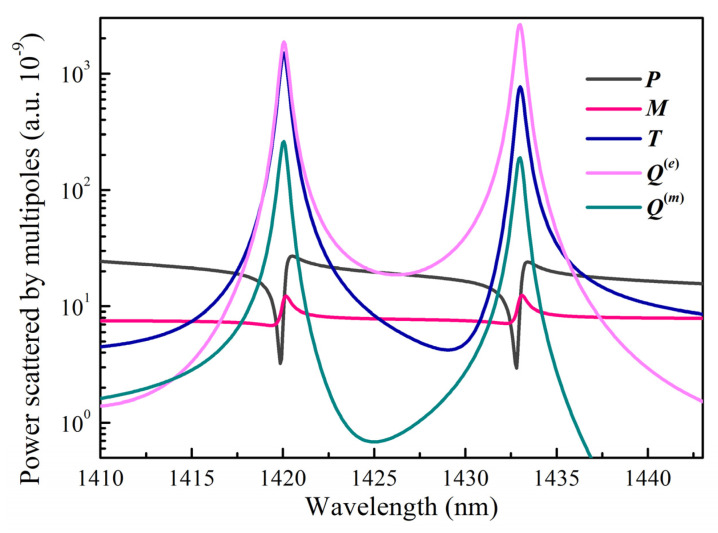
Far−field scattering contributions of electric dipole P, magnetic dipole M, toroidal dipole T, electric quadrupole $Q^{\left(e\right)}$, and magnetic quadrupole $Q^{\left(m\right)}$.

**Figure 6 nanomaterials-13-00505-f006:**
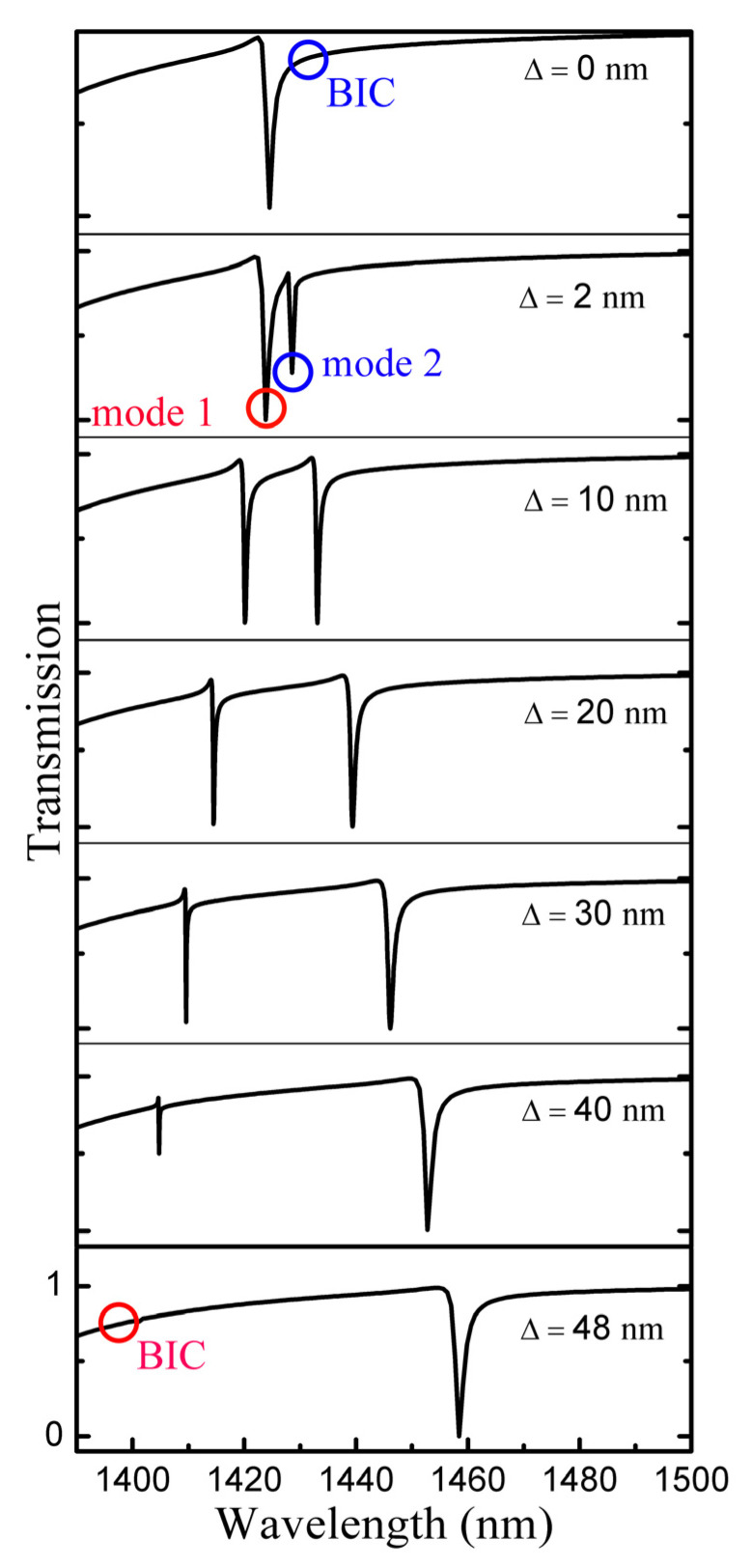
The calculated transmittance spectra when Δ changes from 0 nm to 48 nm. Two resonance responses are marked by mode 1 and 2, respectively.

**Figure 7 nanomaterials-13-00505-f007:**
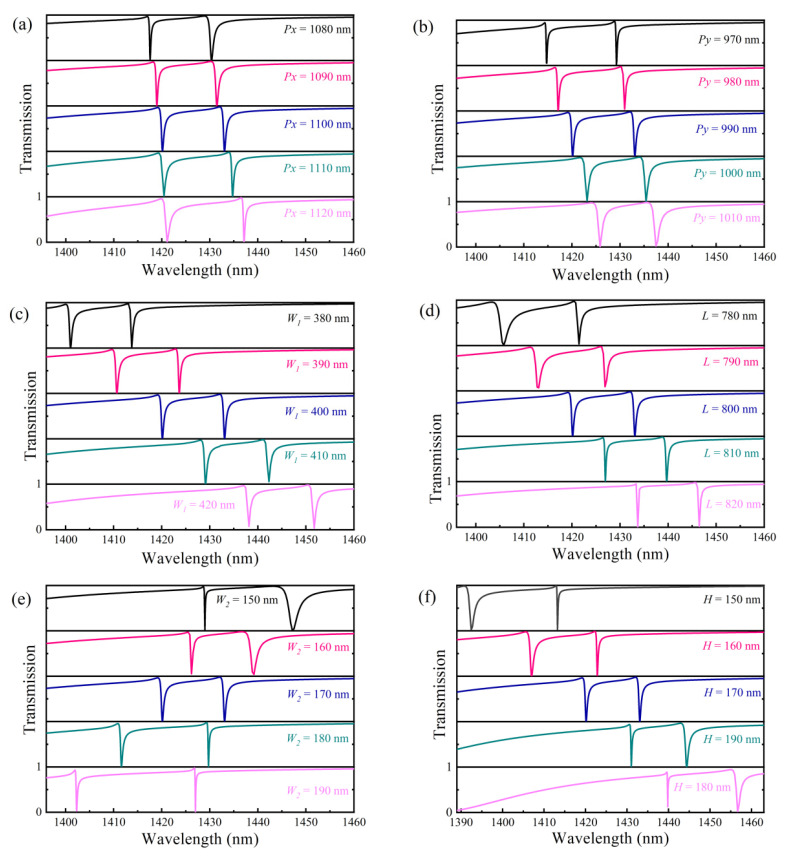
The influence of structural parameters on resonance characteristics. The influence of (**a**) the period *Px*, (**b**) *Py*, (**c**) *W*_1_, (**d**) *L*, (**e**) *W*_2_, and (**f**) the height *H* on the resonance peak.

**Figure 8 nanomaterials-13-00505-f008:**
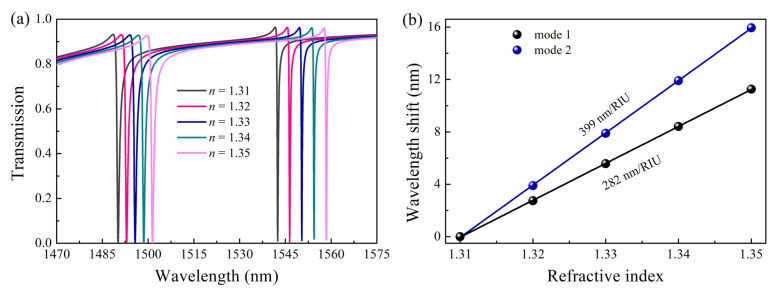
Analysis of sensing performance. (**a**) Variation of transmission spectra for different analyte’s refractive indices; (**b**) Refractive index sensitivity of resonance.

**Table 1 nanomaterials-13-00505-t001:** Comparison of sensitivity and FOM for various sensors.

Reference Year	Material	Structure	Sensitivity (nm/RIU)	FOM (RIU^−1^)
[48] 2017 (Exprement)	Au	grating	470	31
[49] 2018 (Simulation)	TiO_2_/SiO_2_/Ag	Fiber	2610	168
[50] 2019 (Simulation)	Si_3_N_4_, SiO_2_	Grating-waveguide	110	190
[13] 2020 (Exprement)	SiO_2_	hollow cuboids	161	80
[14] 2021 (Simulation)	Si	Lucky knot	986	32.7
[51] 2022 (Simulation)	Au	Split ring-dumbbell	445	1.5
This work (Simulation)	Si	Square hole	399	1281

## Data Availability

The data that support the findings of this study are available from the corresponding author upon reasonable request.

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
