# Peer review of "High-Sensitivity Sensing in All-Dielectric Metasurface Driven by Quasi-Bound States in the Continuum"

_nanomaterials, 2023, doi:10.3390/nano13030505_

Round 1

Reviewer 1 Report

               The article “High-sensitivity Sensing in All-dielectric Metasurface Driven by Quasi-Bound States in the Continuum” by Jing et al presents the theory and simulation work of an all-dielectric metasurface refractive index sensor composed of asymmetric square nanohole arrays that exhibit torodial dipole and electric quadruple resonance modes in the near-infrared region. Authors did a solid job in presenting a nanostrutural design that is convenient and practical to fabricate and following up with thorough analysis on resonance performance, transmission performance, and refractive index with respect to structural parameters. The results are convincing that the metasurface structure in future can offer ultrasensitive sensing capabilities for various engineering applications. I would like to address several issues to modify the paper.

1.      There is a spelling mistake in title, change “Contunuum” to “Continuum”

2.      In introduction section line 71. Please follow up with explanations on why we need “studies on the design of highly sensitive refractive index sensors on silicon metasurfaces on glass substrates.”

3.      Line 112, change “more and more wide” to “wider”

Reviewer 2 Report

The authors introduced a quasi-BIC metasurface with multipole decomposition analysis. There are still several questions need to be answered:

  1. In the introduction, the authors state “BICs have two types, symmetry-protected BIC and Friedrich-Wintgen BIC (FW-BIC) or accidental BIC [17]” To my knowledge, the Friiedrich-Wintgen BIC is one type of accidental BIC and Febry-Perot BIC is also a type of accidental BIC(10.1038/natrevmats.2016.48). The authors might need to further verify the sentence to avoid misunderstanding.

  2. The relationship between the asymmetry parameter and the fano resonance quality factors follows the inverse quadratic law. It is suggested that the authors could draw the Q vs asymmetric parameter to further verify its symmetry-protected origin in Figure 2.

  3. How can we link the surface current distribution in Figure 3 with the multipole decomposition in Figure 4? The authors state “Two current loops circulating in the opposite directions yield a superposition of a magnetic quadrupole (MQ) and a TD”, I can understand the TD part but the MQ part is confusing. The authors might need to give more explanation on that.

  4. It is better to include the BIC-state responses in addition to the quasi-BIC states to prove its BIC-related responses in Figure 5.

  5. Why “Compared with other complex nanostructures, the proposed metasurface is more feasible and practical, which may open up an avenue for the development of ultrasensitive sensors.”? The authors might need to clarify the sentence with convincing evidence. The authors need to also include other published work and compare some critical values with them to emphasize the importance of this design.

Reviewer 3 Report

Comments on the manuscript

In this manuscript entitled “High-sensitivity sensing in all-dielectric metasurface driven by quasi-bound states in the continuum,” the authors presented a systematic study on all-dielectric metasurfaces supporting multiple high Q resonances, which is driven by the physics of asymmetric bound states in the continuum (BICs). They also included index sensing simulation results for two quasi-BIC modes. Overall, the manuscript, in general, is interesting and their simulation results are of interest to the field of dielectric metasurfaces. Also, the manuscript is technically sound with well-supported conclusions and assertions.

Thus, this manuscript meets the scope of nanomaterials. I recommend publishing this work after the following comments are addressed.

1.      My major concern is the motivation of the manuscript. The unit-cell structure design is new. However, it is very similar to recent experimental work by Andrey Miroshnichenko and co-workers [Andrey Miroshnichenko et al. "Bound States in the Continuum in Asymmetric Dielectric Metasurfaces." Laser & Photonics Reviews: 2200564]. This is a simulation-only work. Thus, why is the “asymmetric BIC + sensing” structure novel? What are your unique advantages compared with previous studies? Please comment.

2.      “In the near-infrared range, metasurfaces based on metal surface plasmon structures exhibit low quality factors (Q factor) due to the ohmic loss of metal, resulting in poor detection performance [8,9].” According to my knowledge, plasmonic metasurfaces or nanostructures have better light confinement than their dielectric counterparts and thus exhibit superior performance in biosensing and quantum technologies. Dielectric and plasmonic platforms both have their unique advantages for sensing applications in various scenarios depending on different requirements. Thus, this claim is not very good.

3.      “However, there are limited studies on the design of highly sensitive refractive index sensors on silicon metasurfaces on glass substrates.” It is meaningless to claim this point since there is no justification as to why one would want to conduct refractive index sensing experiments in silicon metasurfaces on glass substrates, which is technologically many choices for index sensing using many other platforms, such as individual metallic nanoparticles and dielectric whispering gallery resonators, to name just a few. Hence, why so sudden?

4.      “The resonance peak of the quasi-BIC mode is larger, and its Q-factor is smaller.” Usually, “larger” is not the right word to describe resonance linewidth broadening.

5.      “The idea of bound states in the continuum (BIC) has been included into various metasurface designs in order to achieve higher Q resonances [16-18]” Correct, but some related plasmonic BICs works are not included. For example (1) BICs in anisotropic plasmonic metasurfaces; (2) multiple resonances BICs, please see [Liang, Yao, et al. "Hybrid anisotropic plasmonic metasurfaces with multiple resonances of focused light beams." Nano Letters 21.20 (2021): 8917-8923]. Better to include these two aspects to provide a more comprehensive picture.

6.      For multipolar decomposition shown in Figure 4. Although the near field distribution shows evidence for EQ and TD (figure 3). However, how do different ED/ EQ/MQ/MD/TD components contribute to the high-Q quasi-BIC resonances?

7.      “Compared to metal or hybrid metal-dielectric sensors, the FOM are higher.” Please provide evidence to support your claim. 

Round 2

Reviewer 2 Report

The manuscript is suggested to be published with the current version.